# Effect of Carbon Nanofiber Distribution on Mechanical Properties of Injection-Molded Aramid-Fiber-Reinforced Polypropylene

**DOI:** 10.3390/polym16081110

**Published:** 2024-04-16

**Authors:** Tetsuo Takayama, Shunsuke Kobayashi, Yuuki Yuasa, Quan Jiang

**Affiliations:** Graduate School of Organic Materials Science, Yamagata University, Yonezawa 992-8510, Japan

**Keywords:** organic fiber, injection molding, whisker, reinforcement, interaction force

## Abstract

The mechanical recycling of discarded plastic products as resources for environmental preservation has recently gained research attention. In this context, it is necessary to use waste materials for fiber-reinforced thermoplastics (FRTP). Glass and carbon fibers are often damaged by shear and compression during melt-forming processes. To achieve a sustainable society, it is necessary for thermal recycling to produce minimal to no residue and for mechanical recycling to maintain the length of fibers used in FRTP to preserve their performance as a reinforcing agent. Aramid fibers (AFs) do not shorten during the melt-molding process, and their composites have excellent impact strength. On the other hand, plastics reinforced with glass or carbon fibers are reported to have a superior strength and modulus of elasticity compared to aramid fibers. This study investigates the dispersion of a carbon nanofiber (CNF), a whisker, as the third component in aramid-fiber-reinforced polypropylene (PP/AF). The results and discussion sections demonstrate how the dispersion of CNF in PP/AF can enhance the mechanical properties of injection-molded products without compromising their impact resistance. The proposed composition will have excellent material recyclability and initial mechanical properties compared to glass-fiber-reinforced thermoplastics.

## 1. Introduction

Fiber-reinforced thermoplastic (FRTP) is a composite material made by combining fibers with thermoplastic (TP). It has a higher modulus of elasticity and strength than TP alone and can be molded and processed using the same methods as TP [1]. Due to its high specific strength and modulus of elasticity compared to metals and ceramics, it is commonly used in applications where weight reduction is necessary [2,3].

The use of discarded plastic products as resources has gained attention in the field of environmental conservation [4]. This is especially true for daily necessities and automobiles, where waste is abundant [5,6,7]. There are currently three methods for waste utilization: thermal recycling, chemical recycling, and mechanical recycling. It is important to note that waste utilization methods have different environmental impacts and should be carefully considered. Chemical recycling decomposes waste and refines it to monomers, which are then polymerized again. Mechanical recycling involves remolding waste. Thermal recycling is the most common method used for plastics, but it has the drawback of increasing carbon dioxide emissions and cannot be repeated. From an objective standpoint, thermal recycling may not be the most suitable method for utilizing plastics in a sustainable society. Therefore, it is crucial to promote chemical and mechanical recycling. Mechanical recycling, which involves remolding waste materials, has the lowest environmental impact of the three recycling methods.

In the case of FRTPs, it is necessary to use waste materials. The main reinforcement materials used in FRTPs are glass and carbon fibers [8,9]. These fibers are inherently brittle and are often reported to fail due to shear and compression during the melt-forming process [10,11]. These fibers are left as residue when thermal recycling is performed. Mechanical recycling involves a melt-forming process that shortens the fibers, compromising their mechanical properties. Thus, the fibers should not shorten during this process. To achieve a sustainable society, FRTP fibers must retain their performance as reinforcing agents and leave minimal residue when thermally recycled.

An aramid fiber is a linear polyamide that does not shorten during the melt-forming process. It has both excellent strength and toughness, making it a popular choice for bulletproof vests. Despite being more expensive than glass fibers, aramid fibers are considered cost-competitive in terms of mechanical recycling. Therefore, aramid fibers were chosen as the reinforcing fibers for this study.

This study focuses on thermoplastics reinforced with aramid fibers. According to previous research [12,13], this material has excellent impact strength. However, it is also reported to have inferior strength and modulus of elasticity compared to plastics reinforced with glass or carbon fibers. One reason for this is that aramid fibers are too flexible and are dispersed in a deflected state inside the molded product. It has been noted that the mechanical properties of aramid fibers vary significantly between their axial and radial directions, which may lead to inadequate reinforcement. To use aramid fibers as a substitute for glass or carbon fibers, it is necessary to take measures to increase their strength and modulus of elasticity.

This study examines the dispersion of a third inorganic filler in aramid-fiber-reinforced plastics. The filler’s material recyclability is also taken into account. Specifically, we focus on whiskers, which are submicron fibers with an aspect ratio of approximately 10 (length to diameter ratio). Due to their small size, whiskers have few internal defects and are known to have a strength close to the ideal strength value [14,15]. As whiskers are shorter than conventional fibers, the possibility of fiber shortening due to shear stresses generated during the melt-forming process is small. Therefore, we consider whiskers to be a filler with excellent material recyclability. In this study, carbon nanofibers were used among the whiskers.

Compared to carbon nanotubes and cellulose nanofibers, carbon nanofibers have a smaller aspect ratio. However, they exhibit excellent dispersibility and a high reinforcing effect. As a result, they were chosen as the third component with the expectation that the desired effect could be achieved with a small nanofiber content, regardless of the molding process. There are few studies that use carbon nanofibers as a filler. A comprehensive study on this topic could potentially lead to the proposal of development and design guidelines for FRTP, with a focus on material recyclability.

The aim of this study is to investigate whether the mechanical properties of aramid-fiber-reinforced thermoplastics can be enhanced by dispersing carbon nanofibers. To achieve this, injection-molded parts of the aforementioned materials were prepared, and their interfacial mechanical properties and notched impact strength were evaluated. The aim of this study was to quantitatively determine the effect of carbon nanofiber dispersion on the mechanical properties of aramid-fiber-reinforced thermoplastic injection-molded products by comparing the results obtained from different compositions.

## 2. Materials and Methods

### 2.1. Materials

Table 1 displays the raw materials used in this paper, including polypropylene as the base material, chopped aramid fibers with a fiber length of 1.0 mm and a fiber diameter of 12 μm, chopped glass fiber with a fiber length of 3.0 mm and a fiber diameter of 13 μm, and two types of carbon nanofibers with different fiber lengths (S-CNF for the relatively short fiber length and L-CNF for the longer fiber length).

### 2.2. Melt-Blending

Table 2 displays the compositions analyzed in this paper. The composite materials were produced by melting and kneading the raw materials in a twin-screw melt-extrusion kneader (IMC-00 type, L/D = 25; Imoto Machinery Co., Ltd.) with a 15 mm screw diameter at a fixed temperature of 230 °C and a screw speed of 60 rpm.

### 2.3. Injection Molding

The composite pellets obtained through melt kneading were injected into a molding machine (C, Mobile0813; Shinko Sellbic Co., Ltd., Tokyo, Japan) to produce test articles. The injection molding conditions are detailed in Table 2, including injection molding temperature (T_inj_), mold temperature (T_mold_), injection speed (V_inj_), holding pressure (P_hold_), filling holding time (t_inj_), and cooling time (t_cool_). The injection speed was set at 30 mm/min, while the injection molding and mold temperatures were set at 230 °C and 50 °C, respectively. The holding pressure was adjusted to ensure that a high-quality product was produced. Figure 1 displays the molded product, which has a 5 mm width, 2 mm thickness, and 50 mm length. The mold can be modified to alter the flow path and create a weld at the center of the product. For this study, two types of molded products were prepared: one without welds and one with welds.

### 2.4. Three-Point Bending Test

Molded parts without welding, obtained through injection molding, underwent three-point bending tests following ISO 178 [16]. The product was tested flatwise, with a loading speed of 10 mm/min and a 40 mm distance between fulcrums, using a tabletop tensile and compression testing machine (MCT-2150, A&D Co., Ltd., Tokyo, Japan) with a 500 N load cell capacity. The flexural stress–flexural strain curve was created using Equations (1) and (2) to determine the flexural stress σ_f_ and strain ε_f_ from the obtained load–deflection curve.
(1)$$\sigma_{f}=\frac{3PS}{2WB^{2}}$$
(2)$$\varepsilon_{f}=\frac{6\delta B}{S^{2}}$$
where B is the thickness of the molded product, W is the width of the molded product, S is the distance between the fulcrums, P is the load, and δ is the deflection. From the obtained flexural stress–flexural strain curve, the maximum flexural stress was evaluated as the flexural strength σ_fS_ and the slope at the beginning of loading as the flexural modulus E_f_, respectively. Five tests were conducted for each composition, and the average value was evaluated as the characteristic value.

### 2.5. Uniaxial Tensile Test

The ISO 527 standard was followed to perform uniaxial tensile tests on injection-molded weld-formed parts [17]. The test was conducted using a compact universal mechanical testing machine (FDA-1kN, IMADA Co., Ltd., Aichi, Japan) with a loading speed of 1.5 mm/min and a chuck-to-chuck distance of 22.5 mm. The obtained load–displacement curve was used to determine the nominal tensile stress and true strain using Equations (3) and (4), respectively.
(3)$$\sigma_{n}=\frac{P}{WB}$$
(4)$$\varepsilon_{t}=\ln\left(1+\frac{\Delta }{L}\right)$$
where L and Δ denote the distance and displacement between chucks, respectively. The test was conducted five times for each composition, and the average value was evaluated as the characteristic value.

### 2.6. Short-Beam Shear Test

Short-beam shear tests were conducted on injection-molded parts with weld formation [18]. The test was performed using a tabletop tensile and compression testing machine (MCT-2150, A&D Co., Ltd., Tokyo, Japan) with a load cell capacity of 500 N. The molded product was installed edgewise, and the test was carried out with a loading speed of 10 mm/min and a 10 mm distance between fulcrums. The rigidity and average shear stress were calculated from the load–deflection curve. Rigidity was obtained by differentiating the load by deflection. The average shear stress was obtained using Equation (5). The resulting data can be plotted on a rigidity-averaged shear stress curve.
(5)$$\tau =\frac{3P}{4BW}$$

The product being studied is made of short-fiber-reinforced thermoplastic that contains discontinuous fibers. It is anticipated that most of the fibers will be oriented at an angle to the flow direction. When the orientation is oblique, interface slippage is expected to occur at low shear stresses in the direction of orientation, which is close to parallel to the loading direction. During the test, the specimen’s rigidity is expected to decrease suddenly due to slippage. Shear stress is conjugate and perpendicular to the loading direction, and there are two orthogonal directions. If the fibers are three-dimensionally oriented, slip is expected to occur at the interface in all three directions due to shear stress. In short-beam shear tests performed on moldings without welds, three points of discontinuous rigidity reduction are expected to occur. To analyze this, the study utilized weld regions where fibers are dispersed in a two-dimensional oriented state. Figure 2 displays an example of the rigidity-averaged shear stress curve obtained, which shows two points of a discontinuous decrease in rigidity, as expected. Slippage occurs at the interface between the fiber and the matrix. The average shear stresses at these points are τ_1_ and τ_2_, respectively. To determine the most frequent value of the fiber orientation angle, θ, Equation (6) was used.
(6)$$\theta =\tan^{-1}\frac{\tau_{2}}{\tau_{1}}$$

Using the obtained θ and τ_1_, the interfacial shear strength (IFSS) τ_I_ was determined using Equation (7).
(7)$$\tau_{I}=\frac{\tau_{1}}{\cos\theta }$$

Tests were conducted 10 times for each composition, and the average value was evaluated as the characteristic value.

### 2.7. Notched Charpy Impact Test

The molded product was tested using the notched Charpy impact test in accordance with ISO 179-1 [19]. The test was performed flatwise with a loading velocity of 2.91 m/s and a distance of 40 mm between fulcrums. The notches were machined to approximately 1 mm in width and had a radius of curvature of 0.25 mm. The swing angle was used to determine the impact absorption energy U. The notched Charpy impact strength a_iN_ was then calculated using Equation (8).
(8)$$a_{iN}=\frac{U}{B\left(W-c\right)}$$
where c is the notch depth. The test was conducted five times for each composition, and the average value was evaluated as the characteristic value.

### 2.8. X-ray CT Imaging

The injection-molded moldings without welding were observed for fiber orientation using an X-ray CT system (ScanXmate-D225RSS270, Comscantecno Co., Ltd., Yokohama, Japan). The observed areas are presented in Figure 3. The surface area of the molded product was observed as the area related to a three-point bending, and the core layer was observed as the area related to the Charpy impact. An example of the evaluated images is shown in Figure 4. Because aramid fibers and polypropylene have a small difference in density, it is challenging to obtain a clear contrast in the original image. To facilitate subsequent image analysis, image processing was performed to create a strong contrast, as shown in Figure 4b. The observed results were used to determine the fiber orientation angle with respect to flow direction using image analysis software (WinROOF2023 ver.6.5.0; Mitani Corp., Fukui, Japan), and the average value was evaluated as the characteristic value. Assuming that the fibers were oriented at the angle of the line connecting the start and end points, the angle was determined when the fibers were dispersed in a deflected state.

### 2.9. Fracture Surface Observation

The fracture surface obtained from the Charpy impact test of the molded product was observed using a digital microscope (TG200HD2-Me, Shodensha Co., Ltd., Osaka, Japan). Image analysis software (WinROOF2023 ver.6.5.0; Mitani Corp., Fukui, Japan) was used to extract more than 500 pulled fibers from the observed images, and the average length and standard deviation of the fibers were determined. 

## 3. Results

### 3.1. Flexural Properties

Figure 5 displays the load–deflection curves obtained from the three-point bending test. The compositions analyzed in this paper did not fracture under three-point bending loading. PP/AF compositions exhibited a smaller slope and lower maximum load at the initial stage of loading compared to PP/GF. The flexural strength and flexural modulus obtained from the three-point bending test results are presented in Figure 6. Both figures show the results for PP/GF and CNF content on the horizontal axis. The addition of 3 wt% or more of CNF in PP resulted in a higher value than that of PP/GF. There was no significant difference in the improvement of flexural strength with CNF fiber length, but the greatest improvement in the flexural modulus was observed when 5 wt% of L-CNF was added.

These results suggest that incorporating an appropriate amount of S-CNF or L-CNF into a system of AF dispersed in PP can enhance flexural strength and flexural modulus.

### 3.2. Interfacial Mechanical Properties

Figure 7 shows the rigidity–load curves obtained from the short-beam shear test. The figure displays arrows that indicate points of discontinuous decrease in rigidity, and these points were used to determine the interfacial shear strength according to the method described in 2.6. The rigidity of PP/AF is lower than that of PP/GF at the same load. The rigidity increases with the amount of S-CNF or L-CNF dispersed in PP/AF. Moreover, dispersion of S-CNF or L-CNF at 3 wt% or higher resulted in higher rigidity than PP/GF. Finally, the dispersion of S-CNF or L-CNF tended to shift the point of discontinuous decrease in rigidity to the lower load side. The presented trend is not correlated with the content of S-CNF or L-CNF. Figure 8 displays the evaluation results of interfacial shear strength obtained from the short-beam shear test. The figure displays standard deviations through error bars, and the same results for PP/GF are shown with a dotted line. The figure shows that the IFSS of PP/AF is slightly lower than that of PP/GF, indicating that the dispersion of CNF in PP/AF tends to slightly decrease IFSS.

Figure 9 shows the nominal stress–true strain curves obtained from uniaxial tensile tests. It can be observed that dispersing 1 wt% of S-CNF or L-CNF in PP/AF reduces the maximum nominal stress. Figure 10 summarizes the maximum nominal stress as weld strength. The dotted line shows the same results for PP/GF. Error bars are shown in the figure but are hidden by the symbols because the standard deviations are very small. The weld strength of PP/AF is higher than that of PP/GF, and the dispersion of CNF in PP/AF decreases weld strength.

The results suggest that the dispersion of S-CNF or L-CNF in a 10 wt% dispersion of AF in PP leads to a reduction in interfacial mechanical properties.

### 3.3. Notched Charpy Impact Strength

Figure 11 shows the Charpy impact strength obtained from the notched Charpy impact test. The figure displays standard deviations through error bars, and the same results for PP/GF are shown with a dotted line. PP/AF exhibits a higher Charpy impact strength than PP/GF, and the same values are comparable to those of PP/AF when S-CNF or L-CNF is dispersed in PP/AF.

In summary, the study found that dispersing an appropriate amount of CNF in PP/AF 10 wt.% can achieve mechanical properties that are at least as good as those of PP/GF 10 wt.%. However, it should be noted that interfacial mechanical properties are reduced.

## 4. Discussions

### 4.1. Carbon Nanofiber Dispersion Effects on Notched Impact Strength of PP/AF Composites

The results of this study indicate that PP/AF has a higher notched impact strength than PP/GF. This is attributed to the energy dissipation mechanism caused by fiber pull-out, as proposed by Quan et al. The notched Charpy impact strength of PP/GF can be calculated using Equation (9) [20], according to Quan et al.’s findings.
(9)$$a_{iN}=\tau_{I}\cos\psi S_{f}l_{p}^{2}$$
where τ_I_ is the interfacial shear strength, ψ is the average fiber orientation angle with respect to the loading direction, S_f_ is the specific surface area at the interface, and l_p_ is the pull-out length. l_p_ is determined using Equation (10) based on the critical fiber length L_c_.
(10)$$l_{p}=\left\{\begin{array}{c} \frac{l}{2}\;\mathrm{for}\;L_{F}<L_{c}\\ \frac{L_{c}}{2}\;\mathrm{for}\;L_{F}>L_{c} \end{array}\right.$$
where L_F_ is the remaining fiber length. L_c_ is obtained using Equation (11) [21].
(11)$$L_{c}=\frac{d\sigma_{FB}}{2\tau_{I}}$$
where d is the fiber diameter and σ_FB_ is the tensile strength of the fiber. As stated in Equation (9), measuring the pull-out length is necessary to quantitatively determine the Charpy impact strength of FRTP. Figure 12 displays the results of a side view of a specimen after a Charpy impact test using a digital microscope. In all compositions, fiber pull-out was observed in most of the crack propagation areas. The length of the pull-out was determined using these images. Figure 13 displays the average pull-out length. The PP/AF pull-through length is approximately 0.3 mm. The length of the fiber in question is less than 0.5 mm, which is half the length of the 1 mm fiber of AF before kneading. This indicates that the remaining fiber length of AF is longer than the critical fiber length. Therefore, we can conclude that l_p_ in the case of PP/AF is L_c_/2. This information is reflected in Equation (9), and the notched impact strength of PP/AF is expressed in Equation (12) below.
(12)$$a_{iN}=\frac{d\sigma_{FB}^{2}V_{f}\cos\psi }{4\tau_{I}}$$

The notched impact strength of PP/AF is governed by five factors, fiber volume content, fiber orientation angle, fiber diameter, fiber tensile strength, and IFSS, as shown in this equation. In the compositions and conditions considered in this study, fiber diameter and fiber tensile strength remain constant, while fiber volume content, orientation angle, and IFSS vary. The structure of the core layer region was observed using microfocus X-ray CT to determine the orientation angle of the fibers. Figure 14 shows the structural observations of the core layer region. The fibers were found to be oriented according to the fountain flow that often occurs in injection molding. No significant changes in this morphology were observed when S-CNFs or L-CNFs were dispersed. The results suggest that there is a small change in the fiber orientation angle and no significant change in volume content. This narrows the focus of the study to IFSS. Figure 7 demonstrates that dispersing S-CNF or L-CNF has little effect on IFSS. This indicates that a trend in notched impact strength was observed when S-CNF or L-CNF was dispersed.

### 4.2. Carbon Nanofiber Dispersion Effects on Flexural Properties of PP/AF Composites

The results of this study indicate that the dispersion of S-CNF or L-CNF in PP/AF can enhance flexural strength and the flexural modulus compared to PP/GF. The mechanisms behind this improvement are discussed in terms of fiber orientation and dispersion state, with a focus on the flexural modulus. To achieve this, the skin layer region’s structure was observed using microfocus X-ray CT. Figure 15 displays the fiber orientation in the skin layer region. The fibers in PP/AF are primarily oriented in the flow direction, but some fibers are dispersed in a curved or bent shape. According to Omidi et al. [22], the improvement in elastic modulus is less significant when fibers are dispersed in a bent or folded orientation. Therefore, the improvement in elastic modulus obtained by dispersing AF was smaller than that obtained by dispersing GF, probably because AF remained longer but was dispersed in a bent and twisted manner. The increase in the flexural modulus of S-CNF or L-CNF can be attributed solely to their reinforcing effect, as fiber bending was not eliminated by these dispersions. The higher flexural modulus of L-CNF compared to S-CNF can be attributed to its higher aspect ratio.

The following section discusses the flexural strength of FRTP in terms of fiber pull-out and interfacial debonding, which are yield conditions for FRTP. The stress required to debond the interface when the fiber orientation angle is φ is given by Equation (13) as follows:(13)$$\sigma_{y}=\frac{\sigma_{w0}}{\sin\varphi }$$
where σ_w0_ is the weld strength. In this study, three-point bending tests were conducted with the specimens placed flatwise. As a result, the deformation in the tensile region can be considered as pure extensional flow deformation with suppressed Poisson’s shrinkage [23]. Equation (14) provides the yield initiation stress obtained in this case.
(14)$$\sigma_{fy}=\sigma_{y}\left(1+\upsilon \right)$$
where υ is Poisson’s ratio. Considering Equations (13) and (14) together, the flexural yield initiation stress σ_fy_ is given by Equation (15) when the yield condition is interfacial debonding.
(15)$$\sigma_{fy}=\frac{\sigma_{w0}\left(1+\upsilon \right)}{\sin\varphi }$$

When full-section yielding occurs in a three-point bending test, there is a 2:3 relationship between the stress at flexural yield initiation and flexural strength. If the flexural yielding condition is interfacial debonding, the flexural strength is mainly determined by the fiber orientation angle and weld strength. The reason for this is discussed based on Takayama’s theory of weld strength, which is expressed in Equation (16) [24].
(16)$$\sigma_{w0}=\left(\alpha_{M}V_{M}-\alpha_{F}V_{F}\right)\left(T_{inj}-T_{test}\right)E_{ct}+\gamma_{i}S$$
where α is the average linear expansion coefficient obtained between the molding and test temperatures, V is the volume content, T_test_ is the test temperature, γ_i_ is the interfacial interaction force, and S is the specific surface area at the interface. The subscripts F and M denote the fiber and matrix phase, respectively. E_ct_ is the modulus of elasticity obtained using the following Equation (17).
(17)$$E_{ct}={\left(\sum_{j=1}^{n}\frac{V_{j}}{E_{j}}\right)}^{-1}$$

Based on Equation (16), it can be inferred that the dispersion of S-CNF or L-CNF affects E_ct_ and the interfacial interaction force, while the content of AF remains relatively constant. Equation (17) indicates that E_ct_ increases with the dispersion of S-CNFs or L-CNFs, while the interfacial interaction force decreases due to the generation of an interface between each CNF and the matrix. Since an interaction force is generated at this interface, it is believed that this force strongly contributes to the exfoliation of the interface between the matrix and AF.

Figure 16 displays the distribution of fiber orientation. The angle shown in Figure 16 is based on the MD direction shown in Figure 15, and the symbol φ indicates the average value of the fiber orientation angle. The dispersion of S-CNF or L-CNF in PP/AF increased the frequency of fibers with an orientation angle between 5 and 35°, and as a result, φ also increased. This suggests that the improvement in bending strength resulting from the dispersion of S-CNF or L-CNF in PP/AF may be due to the fiber orientation angle. However, interfacial debonding is unlikely to occur because the obtained φ is smaller than 45° for all compositions. Based on the discussion above, it can be concluded that the yield condition of the compositions analyzed in this paper does not involve interfacial debonding. Figure 15 illustrates that the fibers in the skin layer were oriented at an angle similar to the flow direction, with some fibers dispersed in a curved or bent shape. This fiber orientation is expected to cause fiber pull-out under tensile loading.

If fiber pull-out is a yield condition resulting from tensile loading, the stress at yield initiation can be determined using Equation (18).
(18)$$\sigma_{y}=\frac{\sigma_{y0}}{\cos^{2}\varphi }$$

In addition, σ_y0_ is obtained using Equation (19).
(19)$$\sigma_{y0}=2\tau_{I}\cos\varphi \frac{l_{p}V_{f}}{dL_{F}}$$
where d is the fiber diameter. Equation (20) expresses the flexural yield initiation stress when the yield condition is fiber pull-out by combining Equations (14) and (18).
(20)$$\sigma_{fy}=\frac{2\tau_{I}\left(1+\upsilon \right)}{\cos\varphi }\frac{l_{p}V_{f}}{dL_{F}}$$

Equation (21) can be obtained by solving for φ as follows.
(21)$$\varphi =\cos^{-1}\left\{\frac{2\tau_{I}\left(1+\upsilon \right)}{\sigma_{fy}}\frac{l_{p}V_{f}}{dL_{F}}\right\}$$

Figure 17 presents a comparison between the calculated φ from Equation (21) and the φ obtained by analyzing the image in Figure 15, expressed as φ_cal_ and φ_exp_, respectively, with ν set to 0.4. The figure displays standard deviations through error bars. In the same figure, dotted lines indicate where the calculated and experimental results agree. The results show good agreement between φ_cal_ and φ_exp_ for PP/GF and PP/AF. This suggests that the model proposed in this paper can be used to analyze the stresses at yield initiation of PP/AF and PP/GF. On the other hand, the value of φ_cal_ for PP/AF with S-CNF or L-CNF dispersed in the composition was higher than φ_exp_. This suggests that the yield condition of PP/AF with S-CNF or L-CNF involves not only AF but also S-CNF or L-CNF, despite the fact that the yield condition is fiber pull-out. Possible factors that may be involved are υ and σ_yo_. The dispersion of S-CNF or L-CNF may increase the specific surface area of the interface, which in turn increases the frictional force required for shear deformation at the interface. This increase in frictional force may lead to an increase in σ_y0_, while the Poisson’s ratio of the CNFs is small, resulting in a decrease in υ. It is believed that the combination of these effects resulted in the flexural strength trends observed in this paper.

The above results and discussion suggest that dispersing carbon nanofibers In aramid fiber-dispersed PP composites can enhance the mechanical properties of injection-molded products without compromising their impact resistance. The mechanical properties obtained were superior to those of glass fiber-dispersed composites. This suggests that organic-fiber-reinforced thermoplastic composites can replace glass-fiber-reinforced thermoplastic composites. The proposed material must have mechanical recyclability in addition to its initial mechanical properties. The mechanical recyclability of FRTP has been confirmed through changes in mechanical properties that occur after repeated extrusion molding [10,11]. As a future development, we plan to investigate changes in the mechanical properties of the molded product caused by multiple extrusion molding cycles to verify whether the composition proposed in this study has excellent mechanical recyclability.

## 5. Conclusions

This paper presents a detailed study of the effect of carbon nanofiber dispersion on the mechanical properties of aramid-fiber-reinforced injection-molded polypropylene. The results obtained are as follows:CNF dispersion in PP/AF improved the flexural strength and flexural modulus without compromising impact resistance. Depending on the CNF content, these values were higher than those of glass-fiber-reinforced injection-molded polypropylene products.Interfacial mechanical properties were slightly decreased by dispersing CNF in PP/AF. This improvement was attributed to the change in interfacial interaction forces caused by the CNF dispersion.The skin layer of PP/AF injection-molded products showed a tendency for fiber orientation to be parallel to the flow direction due to CNF dispersion, while the core layer’s fiber orientation remained unchanged.The impact strength of PP/AF injection-molded products is not dependent on CNF content, which can be explained by changes in AF volume content and IFSS, as predicted by the fiber pull-out model.

In addition to the initial mechanical properties, material recyclability is also required for the materials proposed in this study. As a future development, we plan to investigate changes in the mechanical properties of the molded products resulting from multiple extrusion moldings. This will verify whether the proposed composition has excellent material recyclability.

## Figures and Tables

**Figure 1 polymers-16-01110-f001:**
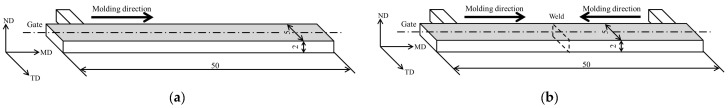
Beam specimen geometries of injection products (**a**) without welding and (**b**) with welding.

**Figure 2 polymers-16-01110-f002:**
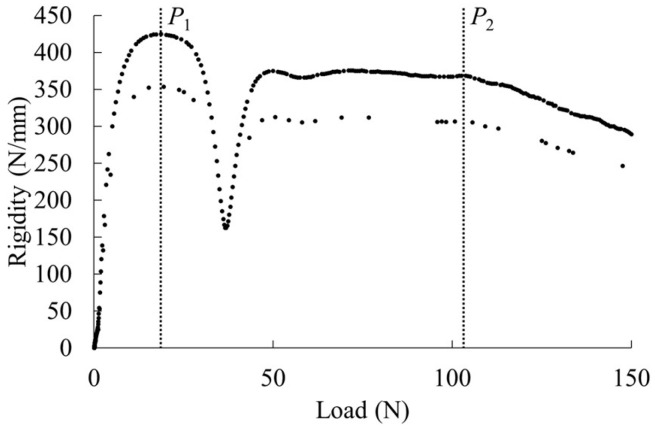
Example of rigidity–load curve. P_1_ and P_2_ indicates the slip points at the interface between the fiber and the matrix.

**Figure 3 polymers-16-01110-f003:**
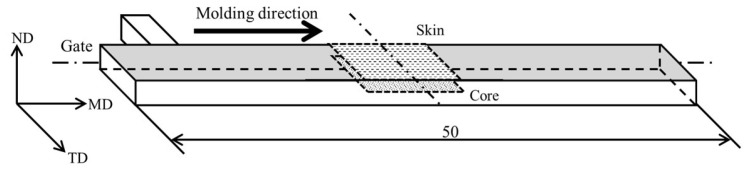
Observation area at fiber orientation using an X-ray CT system.

**Figure 4 polymers-16-01110-f004:**
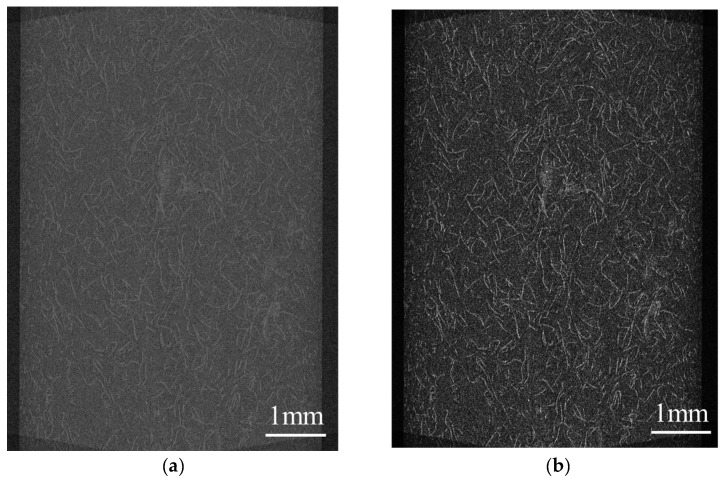
X-ray CT image example of (**a**) original image and (**b**) adjusted image.

**Figure 5 polymers-16-01110-f005:**
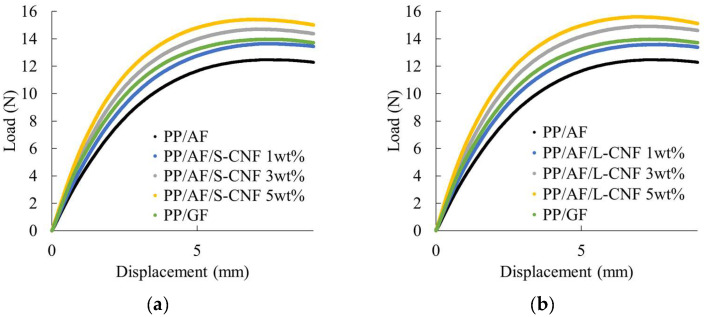
Load–deflection curves of (**a**) PP/AF/S-CNF and (**b**) PP/AF/L-CNF.

**Figure 6 polymers-16-01110-f006:**
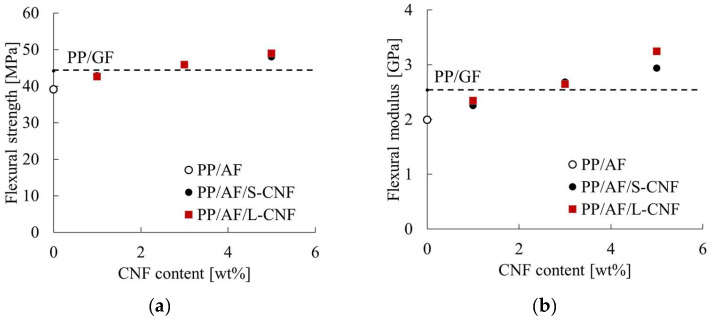
(**a**) Flexural strength and (**b**) flexural modulus of PP/AF/CNF composites.

**Figure 7 polymers-16-01110-f007:**
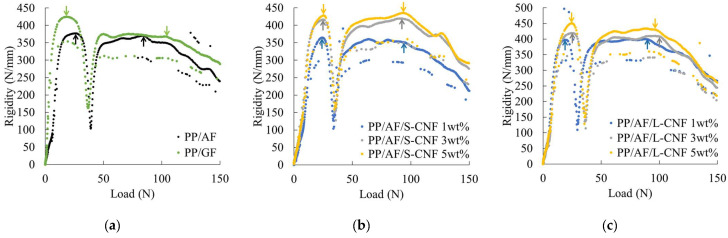
Rigidity–load curves of (**a**) PP/AF and PP/GF, (**b**) PP/AF/S-CNF, and (**c**) PP/AF/L-CNF.

**Figure 8 polymers-16-01110-f008:**
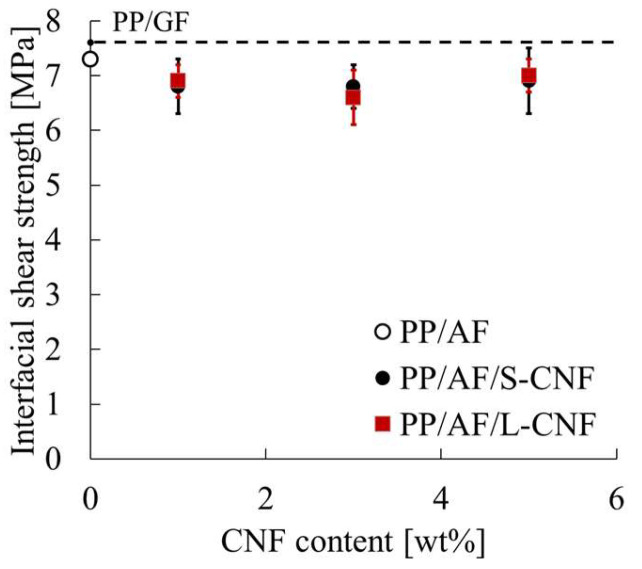
Interfacial shear strength of PP/AF/CNF composites.

**Figure 9 polymers-16-01110-f009:**
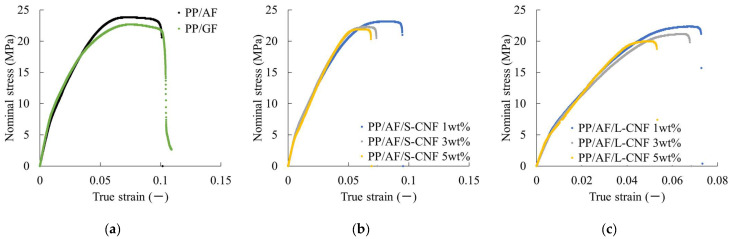
Nominal stress–true strain curves of (**a**) PP/AF and PP/GF, (**b**) PP/AF/S-CNF, and (**c**) PP/AF/L-CNF.

**Figure 10 polymers-16-01110-f010:**
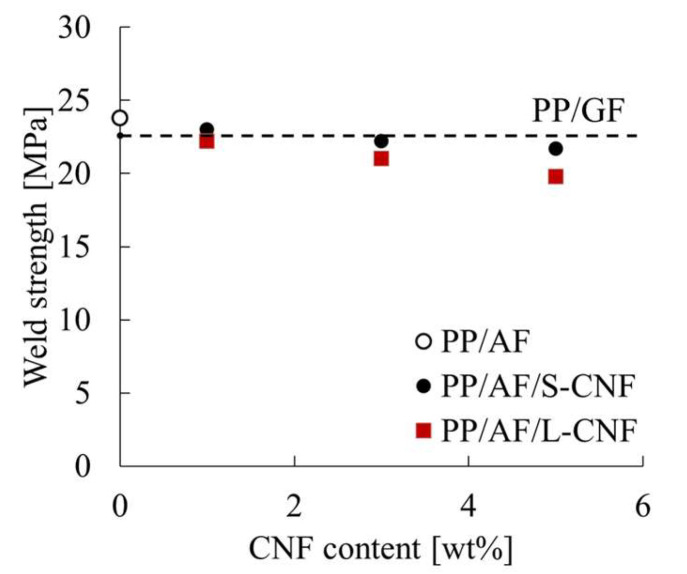
Weld strength of PP/AF/CNF composites.

**Figure 11 polymers-16-01110-f011:**
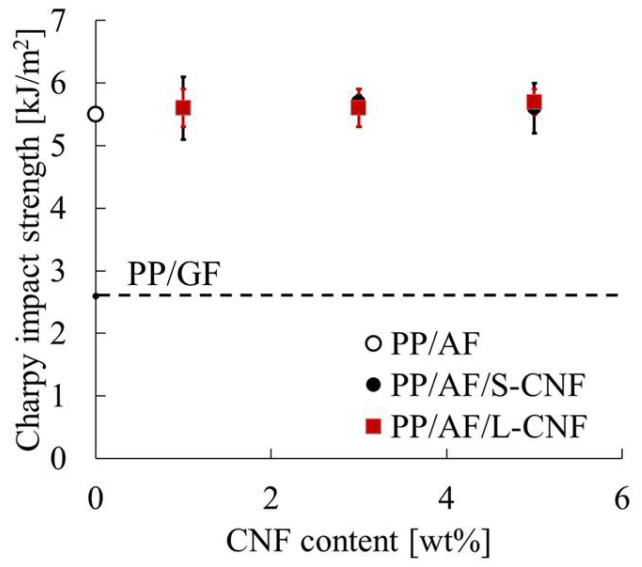
Notched Charpy impact strength of PP/AF/CNF composites.

**Figure 12 polymers-16-01110-f012:**
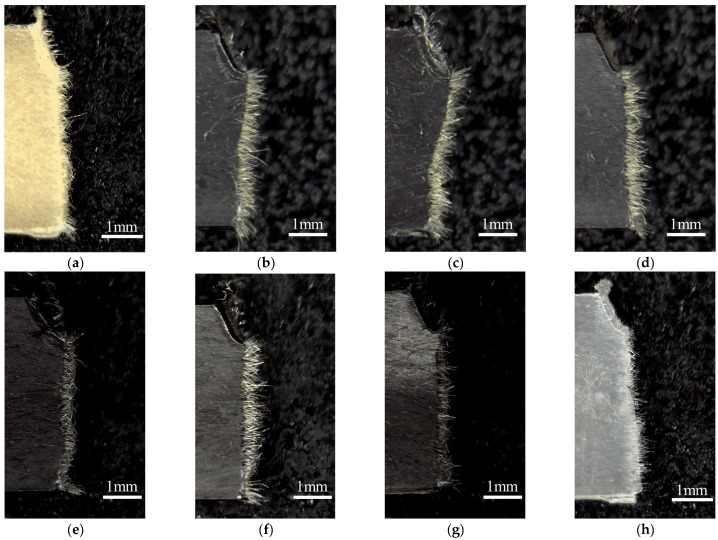
Fracture surface observations obtained from notched Charpy impact tests of (**a**) PP/AF, (**b**) PP/AF/S-CNF 1 wt%, (**c**) PP/AF/S-CNF 3 wt%, (**d**) PP/AF/S-CNF 5 wt%, (**e**) PP/AF/L-CNF 1 wt%, (**f**) PP/AF/L-CNF 3 wt%, (**g**) PP/AF/L-CNF 5 wt%, and (**h**) PP/GF.

**Figure 13 polymers-16-01110-f013:**
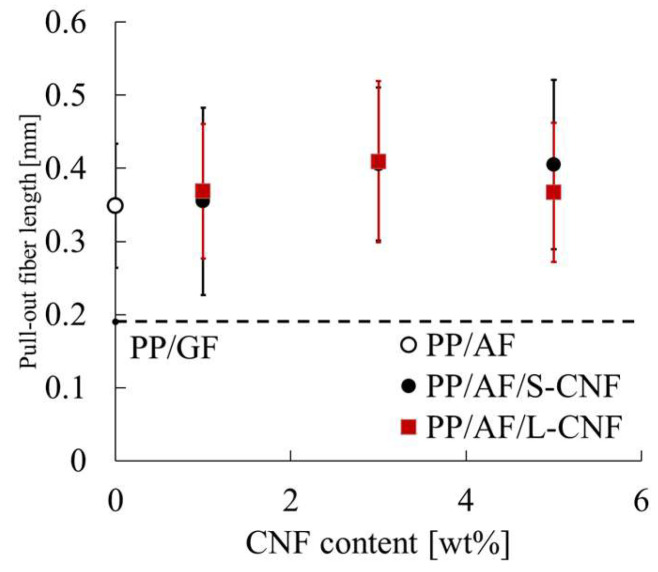
Pull-out fiber length obtained from notched Charpy impact tests.

**Figure 14 polymers-16-01110-f014:**
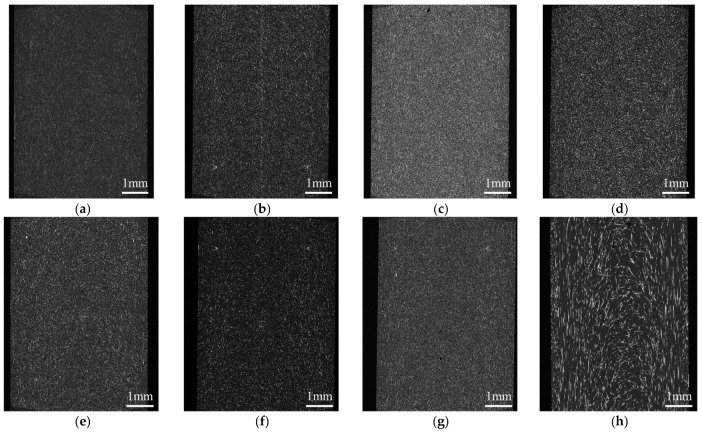
Structural observations of the core layer region of (**a**) PP/AF, (**b**) PP/AF/S-CNF 1 wt%, (**c**) PP/AF/S-CNF 3 wt%, (**d**) PP/AF/S-CNF 5 wt%, (**e**) PP/AF/L-CNF 1 wt%, (**f**) PP/AF/L-CNF 3 wt%, (**g**) PP/AF/L-CNF 5 wt%, and (**h**) PP/GF. The flow direction is from bottom to top.

**Figure 15 polymers-16-01110-f015:**
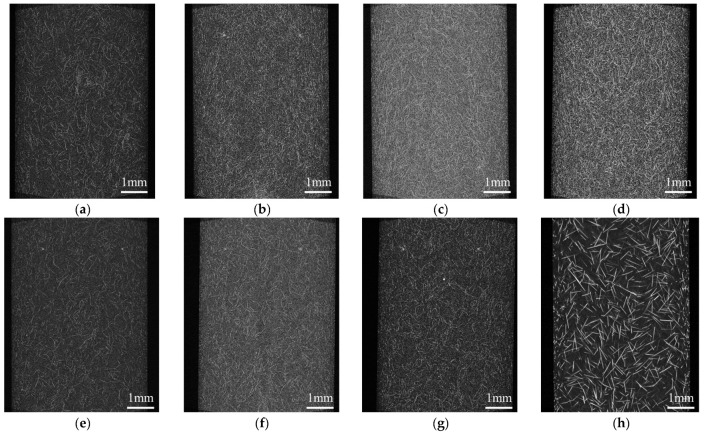
Structural observations of the skin layer region of (**a**) PP/AF, (**b**) PP/AF/S-CNF 1 wt%, (**c**) PP/AF/S-CNF 3 wt%, (**d**) PP/AF/S-CNF 5 wt%, (**e**) PP/AF/L-CNF 1 wt%, (**f**) PP/AF/L-CNF 3 wt%, (**g**) PP/AF/L-CNF 5 wt%, and (**h**) PP/GF. The flow direction is from bottom to top.

**Figure 16 polymers-16-01110-f016:**
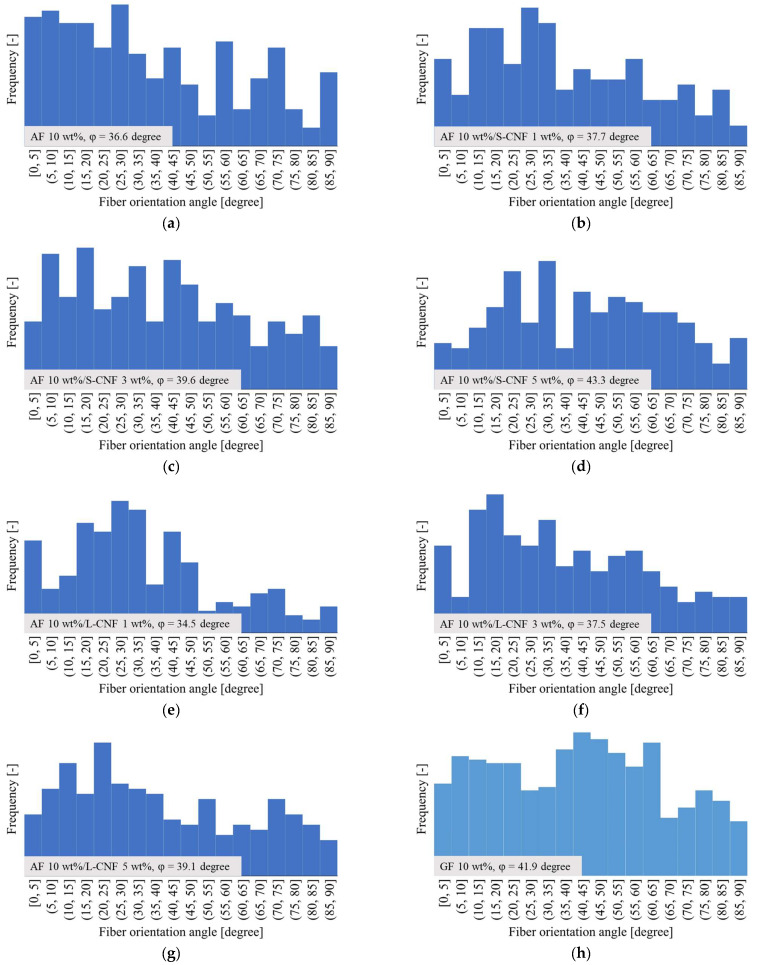
Fiber orientation distributions of the skin layer region of (**a**) PP/AF, (**b**) PP/AF/S-CNF 1 wt%, (**c**) PP/AF/S-CNF 3 wt%, (**d**) PP/AF/S-CNF 5 wt%, (**e**) PP/AF/L-CNF 1 wt%, (**f**) PP/AF/L-CNF 3 wt%, (**g**) PP/AF/L-CNF 5 wt%, and (**h**) PP/GF.

**Figure 17 polymers-16-01110-f017:**
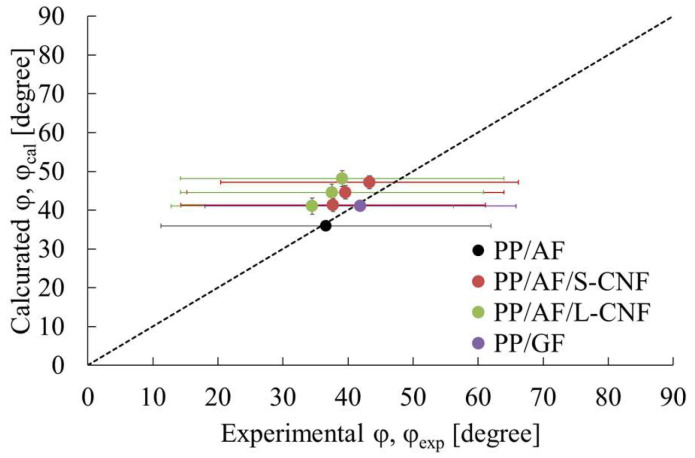
Comparison between the calculated φ from Equation (21) and the φ obtained in the image analysis of Figure 15, expressed as φ_cal_ and φ_exp_.

**Table 1 polymers-16-01110-t001:** Material information.

Material	Code	Manufacturer	Name	Density [g/cm^3^]
Poly propylene	PP	Japan Polypropylene Corp. Tokyo, Japan	Novatec-PP MA1B	0.91
Aramid Fiber	AF	Teijin Co., Ltd. Tokyo, Japan	Technora ZCF1-12	1.4
Glass Fiber	GF	Nippon Electric Glass Co., Ltd. Shiga, Japan	ECS 03 T-351	2.56
Carbon Nano Fiber	S-CNF	Almedio Co., Ltd. Tokyo, Japan	Carbon nanofiber for reinforcement	2.2
	L-CNF	Almedio Co., Ltd. Tokyo, Japan	Carbon nanofiber for electrical conductivity	2.2

**Table 2 polymers-16-01110-t002:** Material composition and injection molding conditions.

PP [wt%]	GF [wt%]	AF [wt%]	S-CNF [wt%]	L-CNF [wt%]	T_inj_ [°C]	T_mold_ [°C]	V_inj_ [m/s]	P_hold_ [MPa]	t_inj_ [s]	t_cool_ [s]
90	10	-	-	-	230	50	10	53	30	15
90	-	10	-	-	230	50	30	56	10	15
89	-	10	1	-	230	50	30	53	10	15
87	-	10	3	-	230	50	30	60	10	15
85	-	10	5	-	230	50	30	63	10	15
89	-	10	-	1	230	50	30	56	10	15
87	-	10	-	3	230	50	30	60	10	15
85	-	10	-	5	230	50	30	63	10	15

## Data Availability

The data presented in this study are available on request from the corresponding author. The data are not publicly available as the data published in the paper are sufficient to understand the claims made in this paper.

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
