# Peer review of "Effect of Carbon Nanofiber Distribution on Mechanical Properties of Injection-Molded Aramid-Fiber-Reinforced Polypropylene"

_polymers, 2024, doi:10.3390/polym16081110_

Round 1
Reviewer 1 Report
Comments and Suggestions for Authors
- Please explain the AF and GF abbreviations in Table 1
- Please provide more information about the twin-screw-extruder used for the materials compounding – producer, length, and diameter of the screws, type of screws, etc.
- Please provide details of the injection molding machine – producer, type of the machine
- Please unify the symbols of flexural strength ( σf or σF) and flexural modulus (εf or Ef)
- What are the P1 and P2 points indicated in Figure 2?
- Please explain the abbreviation IFSS
- In the conclusions the authors claim that “The impact strength of PP/AF injection-molded products is dependent on the CNF content”, however, this is not reflected in the notched Charpy impact strength results presented in Figure 11
Author Response
Thank you for your interest in reviewing this paper. I would like to respond to your suggestions and questions raised below.
Please explain the AF and GF abbreviations in Table 1
- I added the explanations in Table 1.
Please provide more information about the twin-screw-extruder used for the materials compounding – producer, length, and diameter of the screws, type of screws, etc.
- I added the information in text.
Please provide details of the injection molding machine – producer, type of the machine
- I added the information in text.
Please unify the symbols of flexural strength ( σf or σF) and flexural modulus (εf or Ef)
- I changed the subscript for clarity.
What are the P1 and P2 points indicated in Figure 2?
- The explanation of the P1and P2 points were added in caption of Figure 2.
Please explain the abbreviation IFSS
- I added the explanations in text.
In the conclusions the authors claim that “The impact strength of PP/AF injection-molded products is dependent on the CNF content”, however, this is not reflected in the notched Charpy impact strength results presented in Figure 11
- The text has been revised so that the description is appropriate.

Reviewer 2 Report
Comments and Suggestions for Authors
The present manuscript tackles the effect of carbon nanofiber addition on the mechanical properties or aramid fiber reinforced polypropylene. The analysis and characterization are exhaustive, and the topic is well suited for the Polymer journal especially since it covers recyclability aspects related to sustainability. I have some comments on the presentation of the results which should be improved and the same is true for the clarity and reading flow.
) The abstract should focus more on the contribution and novelty of the present work. Currently, only the last sentence places the work into perspective.
) The results of some figures are presented without providing any physical interpretation of the trends. The corresponding discussion should be significantly expanded. This is particularly relevant for Figs. 7-11.
) The effect of the addition of CNF is intriguing as the best performance (for example for the strength and modulus as demonstrated in Fig. 6) is achieved at the highest load. This begs for the question what could happen if the load increases further to values higher than 5wt%. What are the limitations in exceeding the 5wt% value, what would be the maximum achievable load value? Could one expect an optimal value for the performance of the loading?
) Could error bars being added in the data of Fig. 17 ? (in both axes)
) Authors claim that “There have been few studies…” (line 77) but no corresponding references exist.
The manuscript could benefit from a careful editing in the English syntax as there are sentences, especially in the introduction, with very unclear meaning. Some examples:
) It is not clear how the sentence “For these reasons, .. sustainable society” (line 37) is connected to the content of the previous lines.
) “could result in the proposal of development ..” (line 78).
) “GF” (glass fiber) should be properly defined once introduced.
) Holding pressure is denoted differently in the main text (P_h) and in Table 2 (P_hold); “T_cool” should be “t_cool” in Table 2.
) Sentence “To obtain … deflection” (line 145) should be rephrased.
) Some figure legends are written telegraphically. For example, the vertical lines and the P1 and P2 are not properly explained in Fig. 2.
Comments on the Quality of English LanguageThe manuscript is not very clear in some instances and it could benefit from editing in the grammar and syntax.
Author Response
Thank you for your interest in reviewing this paper. I would like to respond to your suggestions and questions raised below.
The abstract should focus more on the contribution and novelty of the present work. Currently, only the last sentence places the work into perspective.
- We have revised the abstract as you indicated.
The results of some figures are presented without providing any physical interpretation of the trends. The corresponding discussion should be significantly expanded. This is particularly relevant for Figs. 7-11.
- Thank you for pointing this out. We have added a description of the figure in the text with as much detail as possible.
The effect of the addition of CNF is intriguing as the best performance (for example for the strength and modulus as demonstrated in Fig. 6) is achieved at the highest load. This begs for the question what could happen if the load increases further to values higher than 5wt%. What are the limitations in exceeding the 5wt% value, what would be the maximum achievable load value? Could one expect an optimal value for the performance of the loading?
- Thank you for your question, we believe that the maximum content of CNF is determined by the molding processability. In the PP/AF studied here, the melt-molding process showed a tendency for CNF dispersion to make molding more difficult. We were not able to quantify this tendency, so we did not add it to the text.
Could error bars being added in the data of Fig. 17 ? (in both axes)
- We can be added error bars in the data of Fig.17 in horizontal axis.
Authors claim that “There have been few studies…” (line 77) but no corresponding references exist.
- Our research did not reveal any examples of the use of carbon nanofibers as filler. We have added "There have been few studies..." to take into account the possibility that there is a lack of research.
The manuscript could benefit from a careful editing in the English syntax as there are sentences, especially in the introduction, with very unclear meaning.
- Thank you for your careful review of this paper. We will respond below to the areas you have pointed out.
It is not clear how the sentence “For these reasons, .. sustainable society” (line 37) is connected to the content of the previous lines.
- We have corrected it to a form that does not cause misunderstanding.
“could result in the proposal of development ..” (line 78).
- We have corrected it to a form that does not cause misunderstanding.
“GF” (glass fiber) should be properly defined once introduced.
- We have added the information of glass fiber in 2.1 Materials.
Holding pressure is denoted differently in the main text (P_h) and in Table 2 (P_hold); “T_cool” should be “t_cool” in Table 2.
- We have corrected it to a form that does not cause misunderstanding.
Sentence “To obtain … deflection” (line 145) should be rephrased.
- We have revised the sentence.
Some figure legends are written telegraphically. For example, the vertical lines and the P1 and P2 are not properly explained in Fig. 2.
- We have corrected it to a form that does not cause misunderstanding.
Comments on the Quality of English Language
The manuscript is not very clear in some instances and it could benefit from editing in the grammar and syntax.
- We have made corrections to the sections you pointed out, and we have also carefully researched the other text and made corrections where appropriate.
